# Research on the NI-MLA Method for Enhancing the Spot Position Detection Accuracy of Quadrant Detectors Under Atmospheric Turbulence

**DOI:** 10.3390/s24206684

**Published:** 2024-10-17

**Authors:** Zuoyu Liu, Shijie Gao, Jiabin Wu, Yunshan Chen, Lie Ma, Xichang Yu, Ximing Wang, Ruipeng Li

**Affiliations:** 1Changchun Institute of Optics, Fine Mechanics and Physics, Chinese Academy of Sciences, Changchun 130033, China; liuzuoyu20@mails.ucas.ac.cn (Z.L.); yiyunsn@ciomp.ac.cn (Y.C.); malie@ciomp.ac.cn (L.M.); yuxichang20@mails.ucas.ac.cn (X.Y.); wangximing@ciomp.ac.cn (X.W.); liruipeng22@mails.ucas.ac.cn (R.L.); 2University of Chinese Academy of Sciences, Beijing 100049, China

**Keywords:** quadrant detector, atmospheric turbulence, non-imaging microlens array, angular measurement characteristic, homogenization

## Abstract

The distorted spots induced by atmospheric turbulence significantly degrade the spot position detection accuracy of the quadrant detector (QD). In this paper, we utilize angular measurement and homogenization characteristics of non-imaging microlens array (NI-MLA) systems, effectively reducing the distortion degree of the spots received on the QD target surface, thereby significantly enhancing the spot detection accuracy of the QD. First, based on the principles of geometric optics and Fourier optics, it is proved that the NI-MLA system possesses the angular measurement characteristic (AMC) within the paraxial region while deriving and verifying the focal length of the system. Then, the QD computation curve characteristics of the system under non-turbulence are explored. This study further elucidates the mathematical principle of the NI-MLA system for mitigating the spot position detection random error of QD (SPDRE-QD) and discusses in depth the relationship between the NI-MLA system’s capability to mitigate the SPDRE-QD and the system’s parameters under various turbulence intensities. Finally, it is experimentally verified that the root-mean-square error (RMSE) of the QD computation values using the NI-MLA system are reduced by a significant improvement of at least 2.44 times and up to 17.36 times compared with that of the conventional optical system of QD (COS-QD) under turbulence conditions ranging from weak to strong.

## 1. Introduction

The quadrant detector (QD), a sensor for spot position based on the principle of the photoelectric effect, is widely utilized as a tracking detector in fields such as laser communication, laser guidance, and quantum key distribution [1,2,3] due to its rapid response, high position resolution, and broad wavelength response range [4]. It is worth noting that ensuring high-precision spot position detection of QD is crucial for maintaining the stability of laser communication and quantum key distribution links, as well as achieving precise strikes in laser guidance. Notably, the high-precision spot position detection of QD can help reduce the pointing errors [5,6], which, in turn, maintains the stability of laser communication and quantum key distribution links and ensures precise targeting in laser guidance, highlighting its significant importance.

However, when a QD is employed in atmospheric channels, turbulence induces random phase changes in the spot [7], and without any intervention, this results in inevitable irregular spot distortions on the QD target surface [3]. This severely impairs the spot position detection accuracy of QD [8], resulting in a further increase of the pointing errors, thereby weakening the system’s tracking stability. Regrettably, the QD’s mere four pixels limit its ability to develop complex algorithms for correcting detection errors caused by distorted spots. Consequently, some researchers have enhanced detection performance by improving QD devices and developing new QD architectures, which reduce the sensitivity of detector response under atmospheric turbulence and the efficiency of photon detection [8,9,10]. However, this has compromised the dynamic range of QD detection, and its applicability is constrained by the intensity of the turbulence, leading researchers to shift their focus from optimizing the QD devices themselves to more effective solutions, namely, employing the active optical system to enhance the spot position detection accuracy of QD in atmospheric turbulence. Specifically, in laser communication, Laguerre–Gaussian beams are used to enhance the spot position detection accuracy of QD under turbulence due to their demonstrated structural stability in atmospheric channels [11,12]. In laser guidance, the structured illumination masks and image reconstruction algorithms are used to reduce angular measurement errors, thereby enhancing the spot position detection accuracy of QD in complex environments such as turbulence [3]. In quantum key distribution, wavefront pre-compensation and received wavefront correction are applied to enhance the tracking stability of the QD system [2]. However, the current active optical systems which can effectively reduce the spot position detection random error of QD (SPDRE-QD) under the influence of turbulence rely on active devices with high power consumption and cost, limiting their widespread application and cross-domain promotion.

Microlens arrays (MLA), with their significant advantages in beam homogenization and shaping, are widely utilized in various fields such as illumination, laser processing, and wavefront sensing [13,14,15]. In this paper, we utilize the homogenization characteristic of non-imaging microlens array (NI-MLA) systems to decouple the distorted spot incident on the system from the spot energy distribution received by the QD target surface, proposing an innovative method of applying the NI-MLA system as the QD optical system. Compared with the conventional optical system of QD (COS-QD), the NI-MLA system significantly enhances the spot position detection accuracy of QD under atmospheric turbulence. Specifically, the QD optical system is essentially an angular measurement system. Within the paraxial range, the spot position is directly proportional to the incident angle of the light. The ratio of the spot position to the incident angle remains constant, equal to the focal length of the system. Therefore, the first issue of this study is to verify the angular measurement characteristic (AMC) of the NI-MLA system in the paraxial region (retaining wavefront tilt) by constructing a NI-MLA system with a certain focal length, both theoretically and experimentally, and then exploring the QD computation curve characteristics corresponding to the NI-MLA system under various parameters. In the core issue of this study, we have developed a mathematical model for the process in which the NI-MLA system transforms distorted spots into relatively uniform spots through “division and overlapped”. This model elucidates the mechanism by which the system mitigates the SPDRE-QD in turbulence. Furthermore, we discuss the trends of how different NI-MLA system parameters affect the system’s capability to mitigate the SPDRE-QD.

The outline of this paper is shown in Figure 1. Specifically, Section 2 briefly analyzes the basic principles of QD spot position detection and simulates the impact of varying turbulence intensities on QD detection. Section 3 and Section 4 analyze, from theoretical and simulation perspectives, the AMC and QD computation curve characteristics of the NI-MLA system (Section 3), as well as its capability to mitigate the turbulence-induced SPDRE-QD (Section 4). In Section 5, the experimental results are provided. Conclusions are drawn in Section 6.

## 2. Analysis of the Atmospheric Turbulence Influence on QD

Before studying the atmospheric turbulence’s influence on QD, the working process of QD is illustrated. Figure 2a illustrates the measurement principles of the quadrant detector (QD). Specifically, part (i) shows the fundamental operating state of the QD with an optical lens placed at the focal plane of the QD, aligning the beam’s angle of incidence with the system’s optical axis to correspond to the QD’s spot position. Part (ii) displays the basic structure of the QD, which consists of four symmetrically arranged p-n junctions around a central point. Each segment of the detector converts the incident light into a proportional current. The beam position is estimated using the “Δ/Σ” formulas, as shown in Equation (1) [11].
(1)$$X=k\sigma_{x}=k\frac{\left(I_{A}+I_{D})-(I_{C}+I_{B}\right)}{I_{A}+I_{B}+I_{C}+I_{D}},\;Y=k\sigma_{y}=k\frac{\left(I_{A}+I_{B})-(I_{C}+I_{D}\right)}{I_{A}+I_{B}+I_{C}+I_{D}}$$
where X and Y represent the actual positions of the light spot in the x and y directions, with $\sigma_{x}$ and $\sigma_{y}$ as their respective calculated values. The k is the coefficient relating the actual position of the light spot to its calculated position. $I_{A}\;\sim {\;I}_{D}$ represent the photocurrents measured in each quadrant. When the spot moves along the x-axis of the QD, the calculation curve of the QD, corresponding to part (iii), is obtained.

Figure 2b presents the simulation results for the QD computation under varying intensities of atmospheric turbulence. The refractive index structure constant ($C_{n}^{2}$) is a crucial parameter signifying the intensity of atmospheric turbulence [16]. In part (i), a Gaussian beam with a 1.2 mm waist radius propagates over a 1 km distance and is received by the COS-QD with a 25 mm focal length, exhibiting energy distributions on the QD surface with $C_{n}^{2}$ ranging from $1\times 10^{-16}$ to $1\times 10^{-12}{\;m}^{-2/3}$. For each $C_{n}^{2}$ value, the QD computation utilizes Monte Carlo simulations to model the spot distribution, as shown in part (ii). Under strong turbulence, the max error approximately reaches 0.52 mm (with an angular measurement error of 20.8 mrad), and the curves lack symmetry and predictability. In part (iii), the curve shows the QD’s computation results as the spot in part (i) shifts along the x-axis on the QD surface. The error curves in the sub-figure are displayed by taking the non-turbulence spot calculation curve as the baseline (red curve) and subtracting the QD computation curves corresponding to different turbulence levels from this baseline to obtain the error distribution. As the value of $C_{n}^{2}$ increases, the spot distortion intensifies, resulting in the QD computation curves progressively deviating from the no-turbulence condition.

It can be seen that turbulence seriously affects the spot position detection accuracy of the QD. Investigating how to cost-effectively and efficiently reduce the degree of spot distortion caused by turbulence to enhance the spot position detection accuracy of the QD has important theoretical research significance and practical application value. Therefore, this study proposes the application of the NI-MLA system in QD optical systems to enhance the spot position detection accuracy of the QD under atmospheric turbulence conditions.

## 3. Verification of the NI-MLA System’s AMC and Analysis of the QD Computation Curve

There are two main types of microlens beam homogenizers: the non-imaging and the imaging MLA (I-MLA) [17]. Because the I-MLA system lacks the AMC within the paraxial region (as demonstrated in the Appendix A), even if the uniformity of the spot generated by the imaging microlens array system (I-MLA) is better than that of the NI-MLA system [18], the I-MLA still cannot be applied in QD optical systems. In this section, the NI-MLA system possesses the AMC within the paraxial region, which will be confirmed by combining geometric optics methods based on theoretical analysis and Fourier optics methods based on simulations. Additionally, a simulation analysis of the QD computation curve characteristics corresponding to the NI-MLA system will be conducted, clarifying the constraints on the parameter selection of the system.

### 3.1. Theoretical Analysis of the AMC

To verify the AMC of the NI-MLA system, the ABCD matrix is utilized in geometric optics to derive the relationship between the height of the image plane ray and the incident angle under paraxial approximation conditions [19]. Also, the focal length analytical formula of the NI-MLA system under general conditions is derived in this subsection. As shown in Figure 3a, the NI-MLA system consists of the MLA with a × a sub-apertures (size *p*, focal length $f_{1}$) and a condenser lens (CL) with a focal length of ${\;f}_{f}$. The distance between CL and MLA is$\;d_{1}$, and the separation between CL and the image plane is $\;d_{2}$. $y_{i\;}$ and ${\;\theta }_{i}$ represent the beam’s incident height and angle, respectively, while $\;y_{o}\;$and ${\;\theta }_{o}\;$ represent the beam’s height of image plane and outgoing angle, respectively, which are collectively represented by the beam vector *O*. The *m* indicates the sequence of MLA sub-apertures (the position of the central sub-aperture is *m* = 1).

Using the ABCD matrix, the beam vector *O* on the image plane of the system shown in Figure 3a can be expressed as:(2)$$\left[\begin{array}{c} y_{o}\\ \theta_{o} \end{array}\right]=\left[\begin{array}{c} \left[d_{2}-d_{1}(\frac{d_{2}}{f_{f}}-1)](\theta_{i}-\frac{y_{i}}{f_{1}})-(\frac{d_{2}}{f_{f}}-1)(y_{i}+mp\right)\\ -(\frac{d_{1}}{f_{f}}-1)(\theta_{i}-\frac{y_{i}}{f_{1}})-\frac{y_{i}+mp}{f_{f}} \end{array}\right]$$

The height of the image plane ray ${\;y}_{o\;}$ and the incident angle ${\;\theta }_{i\;}$ satisfy the following relationship:(3)$$\underset{\Delta \theta_{i}\to 0}{\lim}\frac{\Delta y_{o}}{\Delta \theta_{i}}=d_{2}-d_{1}(\frac{d_{2}}{f_{f}}-1)=F_{NI-MLA}$$

In Equation (3), it can be observed that when the incident angle approaches zero (within the paraxial region), the ratio of the height of the image plane ray ${\;y}_{o}\;$ to the incident angle ${\;\theta }_{i\;}$ remains constant (only related to the distance between MLA and the image plane to CL, as well as the focal length of CL and independent of the parameters of MLA itself). Therefore, this indicates that there is a linear relationship between $\;{\;y}_{o}\;$ and ${\;\theta }_{i\;}$ within the paraxial region, with their ratio representing the focal length *F_NI-MLA_* of the NI-MLA system. In contrast, the I-MLA system does not satisfy this linear relationship (its focal length is calculated to be zero, as detailed in the Appendix A). Above all, from the perspective of geometric optics, it has been proven that the NI-MLA system possesses the AMC within the paraxial region.

When the image plane is located at the focal plane of the CL (i.e.,${\;d}_{2}=f_{f}$), the focal length of the NI-MLA system is $F_{\mathrm{NI}-\mathrm{MLA}}=f_{f}$ according to Equation (3). This result highlights a key characteristic of the NI-MLA system design, that is, under this specific parameter configuration, the focal length of the system is not affected by the MLA position and is always equal to the focal length of the CL. At this point, the spot size *S* is $\;p{\;f}_{f}/f_{1}$ [20]. In practical applications, defocusing the detector appropriately can increase the uniformity of the spot. However, defocusing can also bring changes to the focal length of the system and the spot size of the image plane, as shown in Figure 3b. If the defocus amount is  Δz, i.e., ${\;d}_{2\;}=f_{f}+\Delta z$, then according to Equation (3), the focal length of the NI-MLA system is given by:(4)$$F_{NI-MLA}=f_{f}+\Delta z-d_{1}\Delta z/f_{f}$$

The spot size is $S=p\;[\Delta z/f_{f}+(\Delta z\;+\;f_{f}\;-\;d_{1\;}\Delta z/f_{f})/f_{1}]$.

### 3.2. Simulation Verification of the AMC and Analysis of the QD Computation Curve

In this subsection, Fourier optics theory combined with specific parameters will be employed to conduct numerical simulations of the NI-MLA system’s AMC within the paraxial region while providing numerical solutions for the system’s focal length. In addition, the QD computation curve characteristics corresponding to the NI-MLA system will be thoroughly analyzed, addressing the impact of the diffraction effects of MLA sub-apertures on the QD detection accuracy.

The complex amplitude $U_{0}(x)$ of the plane wave received by the NI-MLA system is $\gamma e^{i\varphi_{1}}(\varphi_{1}=k\;x\;\sin\;\theta )$. After it enters the optical system, the angle spectrum method [19] will be used for transmission simulation to obtain the intensity distribution $e_{\mathrm{NI}-\mathrm{MLA}}$ on the QD target surface in Equation (5).
(5)$$e_{NI-MLA}={\left|F^{-1}\left\{F\left\{F^{-1}\left\{F\left\{\gamma e^{i\varphi_{1}}\cdot e^{i\varphi_{2}}\right\}\cdot H_{1}\right\}\cdot e^{i\varphi_{3}}\right\}\cdot H_{2}\right\}\right|}^{2}$$
where F represent the Fourier transform, $\varphi_{2}$ is the phase of MLA, $\varphi_{3}$ is the phase of CL, and $H_{1}$ and $H_{2}$ are the spatial transfer functions corresponding to distances $d_{1}$ and $d_{2}$ in Figure 3b. The spot energy centroid position ${\mathrm{EC}}_{x}$ can be expressed as:(6)$$EC_{x}=\frac{\sum_{t=1}^{T}\sum_{k=1}^{K}x_{tk}e_{NI-MLA_{tk}}}{\sum_{t=1}^{T}\sum_{k=1}^{K}e_{NI-MLA_{tk}}}=\frac{\sum_{t=1}^{T}\sum_{k=1}^{K}x_{tk}{\left|F^{-1}\left\{F\left\{F^{-1}\left\{F\left\{\gamma e^{ikx\sin\theta }\cdot e^{i\varphi_{2}}\right\}\cdot H_{1}\right\}\cdot e^{i\varphi_{3}}\right\}\cdot H_{2}\right\}\right|}_{tk}^{2}}{\sum_{t=1}^{T}\sum_{k=1}^{K}{\left|F^{-1}\left\{F\left\{F^{-1}\left\{F\left\{\gamma e^{ikx\sin\theta }\cdot e^{i\varphi_{2}}\right\}\cdot H_{1}\right\}\cdot e^{i\varphi_{3}}\right\}\cdot H_{2}\right\}\right|}_{tk}^{2}}$$
where $x_{\mathrm{tk}}$ is the actual position corresponding to each element of the matrixized $e_{\mathrm{NI}-\mathrm{MLA}}$, k=2π/λ. In this paper, *λ* = 633 nm is used for simulation, due to good compatibility of the experimental devices used in Section 5 with this wavelength. Additionally, the visible light band facilitates easier alignment and setup of the optical path. Although this wavelength is different from the actual application scenario (808/1064/1550 nm), the obtained results can still provide valuable reference for practical applications. Regarding the selection of simulation parameters, the field of free space optical communication is taken as an example. When the QD system is in tracking mode, the incident angle residuals received by the optical branch are concentrated near the optical axis of the optical system, which is reflected in the fact that the QD mainly works in a small range around the center of the QD target surface where the typical incident angle residual is 300 μrad. Based on this, ±300 μrad [21] is set as the detection range for the paraxial approximation in this study. However, in order to more comprehensively evaluate the angular measurement performance of the NI-MLA system, the simulation’s incident angle θ range extended to ±500 μrad and the goodness of fit (*R*^2^) is used as an indicator to evaluate the linear relationship between the spot energy centroid position ${\mathrm{EC}}_{x}$ and the incident angle θ. The other parameters for simulation of the AMC of the NI-MLA system are detailed in Table 1.

Before analyzing the AMC of the NI-MLA system, the energy distribution characteristics of the system’s emitted spot are first simulated. As shown in part (i) of Figure 4a, unlike the COS-QD system, which exhibits a circular envelope with an approximately Gaussian energy distribution at defocus positions, the NI-MLA system produces a square envelope with significant diffraction effects at the image plane (∆*z* = 0). In practical applications, the size of the discrete oscillation envelopes containing many diffraction peaks expands with defocusing, and once the defocus is appropriate, a peak-to-valley overlap can be achieved to eliminate the inhomogeneous distribution of the beam [15]. When the defocus amount is Δz =1.5 mm, the spot uniformity (luminous flux contrast) improves from 1.2 as illustrated in part (i) of Figure 4a to 0.6 as shown in part (i) of Figure 4b.

Here, the quantitative analysis results of the corresponding simulation curves of the NI-MLA system in Figure 3 are shown in Table 2.

The part (ii) of Figure 4a,b show the simulation of the relationship between ${\mathrm{EC}}_{x}$ and θ within the range of 1 mrad, and the fitting results correspond to the “NI-MLA system’s AMC” of Table 2. It can be observed that the first-order linear fitting between${\;\mathrm{EC}}_{x\;}$and θ has a goodness of fit *R*^2^ greater than 0.99 at different defocus positions. In addition, the simulated focal length values of the NI-MLA system (i.e., the slope of the fitted curve) are close to the theoretical values calculated using Equation (4), and the error is within 0.9%. Therefore, under the set NI-MLA system parameters, ${\mathrm{EC}}_{x}\;$is directly proportional to θ within the paraxial region, and its proportional coefficient is the system focal length, which is consistent with the theoretical focal length. This preliminarily verifies the conclusion that the NI-MLA system possesses the AMC within the paraxial region (±500 μrad).

Part (iii) of Figure 4a,b shows the QD computation curves of the NI-MLA system at different defocus positions. The simulation results of the curve characteristics (linearity $L_{x}$, sensitivity $k_{x}$, and detection accuracy ${\mathrm{RMSE}}_{x}$) are provided in Table 2. $L_{x}=\delta /y$ where δ is the maximum deviation and *Y* is the test range. $k_{x}$ is equal to the coefficient *k* in Equation (1). ${\mathrm{RMSE}}_{x}$ represents the root-mean-square error of the detection curve. The comprehensive simulation results from the above figure and table indicate that although the diffraction effects at different defocus positions have minimal impact on the AMC of the NI-MLA system, the uneven energy distribution of the spots caused by diffraction effects will negatively impact the spot position detection accuracy of the QD, that is, the QD computation curve of the NI-MLA system at Δz =0 exhibits an overall “S” shape with jagged features locally, and the *RMSE* of the error function (the difference between the simulated values and their central approximation algorithm curve) is 18 μm. However, when the QD position in the NI-MLA system is defocused (Δz =1.5 mm), the QD computation curve of the NI-MLA system becomes smoother (as shown in part (iii) of Figure 4b). As indicated in Table 2, the corresponding error function’s *RMSE* decreases to 4.1 µm, and the curve’s linearity $L_{x}$ and sensitivity $k_{x}$ improve. This indicates that the NI-MLA system can reduce the impact of diffraction effects by adjusting the defocus amount, thereby enhancing the uniformity of the received spot on the QD target surface and, ultimately, enhancing the spot position detection accuracy and linearity of the QD. Additionally, defocusing also reduces the size of the square spot generated by the NI-MLA, further enhancing the QD’s detection sensitivity. In summary, appropriate defocusing in the NI-MLA system can enhance the QD computation curve characteristics. Further discussion on the QD computation curve characteristics under more defocusing parameters of the NI-MLA system, as well as the comparison with the COS-QD’s computation curve characteristics, will be presented in the Experimental Results section.

## 4. Verification of the NI-MLA System Mitigating SPDRE-QD Under Turbulence

In this section, the geometric optics and Fourier optics methods are once again combined to develop a mathematical model of the process by which the NI-MLA system homogenizes distorted spots at the theoretical and simulation levels and explain and preliminarily verify the mechanism of the system’s mitigation of the SPDRE-QD in turbulence.

### 4.1. Theoretical Analysis

As shown in Figure 5, the incident beam is divided into multiple beamlets by the MLA of the NI-MLA system, and the beamlets are overlapped on the focal plane by the condenser lens. In order to verify that the “division and overlapped” process can improve the uniformity of distorted spot after turbulent disturbance, based on the statistical characteristics of spot energy discretization, this study expounds the advantages and specific mechanisms of the NI-MLA system in enhancing the spot position detection accuracy of the QD compared with COS-QD under turbulent conditions.

Laser beam uniformity can be characterized by luminous flux contrast *C* [20]:(7)$$C=\frac{\sqrt{\sum {\left(I_{i}-\bar{I}\right)}^{2}/N}}{\bar{I}}$$
where $I_{i}$ is the energy value of each sample point of the spot, $\overset{-}{I\;}$ is the average value of the sample points, and *N* represents the number of sample points. Essentially, the luminous flux contrast *C* is given by C=σ / μ, which is the coefficient of variation of the matrix. Specifically, it is demonstrated that the NI-MLA system can reduce the luminous flux contrast of the spot energy distribution to illustrate its capability to enhance the uniformity of the spot energy distribution (the smaller value of *C*, the higher the uniformity of the spot). As shown in Figure 5, the intensity distribution of the original incident beam is represented by a matrix *A* (pseudo color image represents the energy distribution of incident turbulent light spot) with a size of Q × Q. The MLA divides *A* into M × M sub-matrices ${\;B}_{k}$, with the size q × q (M=Q/q). The condenser lens then superimposes these sub-matrices $B_{k}$ to form a matrix *O* (intensity distribution of focal plane spot) with the same size of q × q. The distribution of *O* is given by:(8)$$Var(O)=\sum_{k=1}^{M^{2}}Var(B_{k_{ij}})+2\sum_{k=1}^{M^{2}}\sum_{\begin{array}{l} k^{\prime }=1\\ k\neq k^{\prime } \end{array}}^{M}Cov(B_{k_{ij}},B_{k_{ij}^{\prime }})$$
where $B_{k_{\mathrm{ij}}}$ is the element in the *k*-th sub-matrix, corresponding to the element at position (*i*, *j*) in *O*. The luminous flux contrast of the incident spot intensity *A* is given by${\;C}_{A}={\;\sigma }_{A}/\mu_{A}$, and the luminous flux contrast of the focal plane spot intensity *O* of the NI-MLA system is given by $C_{O}=\sigma_{O}/\mu_{O}$. Through the above “division and overlapped” transformation, $\mu_{O}=M^{2}{\;\mu }_{A}$. When the total energy of the spot is fixed, if the light intensity increases in a certain region, the light intensity in other regions may correspondingly decrease, which makes the internal energy distribution of the distorted spot have a correlation, and it also affects the uniformity of the NI-MLA system’s emitted spots${\;C}_{O}$. Next, two possible cases of${\;C}_{O}$ will be discussed. In the first case, when the elements within $B_{k_{\mathrm{ij}}}$ are positively correlated ($\sum_{k=1}^{M^{2}}\sum_{k^{\prime }\neq \;1,\;k\;\neq \;k^{\prime }}^{{\;M\;}^{2}}\mathrm{Cov}\left(B_{k_{\mathrm{ij}}},\;B_{k_{\mathrm{ij}}^{\prime }}\right)>\;0\;\mathrm{in}\;\mathrm{Equation}\left(8\right)$), ${M\;\sigma_{A}\;<\;\sigma }_{O\;}<{M\;}^{2}\;\sigma_{A}$ can be calculated according to Equation (8). Further, the luminous flux contrast of the focal plane spot intensity $O_{\mathrm{ij}}$ is as follows:(9)$$\frac{C_{A}}{M}=\frac{M\sigma_{A}}{M^{2}\mu_{A}}<C_{O}=\frac{\sigma_{O}}{\mu_{O}}<\frac{M^{2}\sigma_{A}}{M^{2}\mu_{A}}=C_{A}$$

This means that $C_{O}$ (the luminous flux contrast of spot $O_{\mathrm{ij}}$) is between $C_{A}/M$ and $C_{A}$. In the second case, when the elements within $B_{k_{\mathrm{ij}}}$ are negatively correlated ($\sum_{\;k\;=1}^{{\;M\;}^{2}}\sum_{k^{\prime }\neq \;1,\;k\;\neq \;k^{\prime }}^{{\;M\;}^{2}}\mathrm{Cov}\left(B_{k_{\mathrm{ij}}},\;B_{k_{\mathrm{ij}}^{\prime }}\right)<\;0$), $\mathrm{Var}(O)\;<{\;M}^{\;2}\mathrm{Var}(A)$ can be calculated. Further, the luminous flux contrast of the focal plane spot intensity $O_{\mathrm{ij}}$ is as follows:(10)$$C_{O}=\frac{\sigma_{O}}{\mu_{O}}<\frac{M\sigma_{A}}{M^{2}\mu_{O}}=\frac{C_{A}}{M}$$

It can be seen that $C_{O}$ is less than $C_{A}/M$, which means that the luminous flux contrast of spot $O_{\mathrm{ij}}$ is higher than the luminous flux contrast of spot *A.*

In summary, in both cases, the uniformity of the spot intensity distribution of the NI-MLA system on the focal plane is better than the intensity distribution of the incident beam, which enhances the consistency of light intensity projected onto each quadrant of the QD and effectively mitigates the impact of spot distortion caused by atmospheric turbulence on the spot position detection accuracy of the QD. Therefore, from the perspective of geometric optics, it has been proven that the NI-MLA system possesses the capability to reduce the SPDRE-QD in atmospheric channels by increasing the uniformity of the distorted spots.

### 4.2. Simulation Verification

In this subsection, Fourier optics theory combined with specific parameters will be employed to simulate and verify the capability of the NI-MLA system to mitigate the SPDRE-QD caused by turbulence and preliminarily analyze the influence of the relative position (∆*z*) of QD in the NI-MLA system on the mitigation effect.

This study employs an atmospheric turbulence model based on the Kolmogorov spectrum. The power spectrum inversion method is applied to calculate the phase screen under different $C_{n}^{2}$ conditions after 1 km of transmission before reaching the optical system. The Gaussian beam undergoes angular spectrum propagation through the phase screens, calculated by segment, to obtain the complex amplitude $\gamma e^{i\varphi_{1}}$ of the distorted wavefront. The emitted light field distribution model of the NI-MLA system is given by Equation (5) where $e_{\mathrm{NI}-\mathrm{MLA}}$ is the light field distribution on the QD target surface after the distorted wavefront $\gamma e^{i\varphi_{1}}$ is modulated by the NI-MLA system, as shown in Figure 6a. In Figure 6b, the COS-QD system receives the same distorted wavefront $\gamma e^{i\varphi_{1}}$, and the output light field distribution at the defocused position $e_{\mathrm{COS}-\mathrm{QD}}$ is given by:(11)$$e_{COS-QD}={\left|F^{-1}\left\{F\left\{\gamma e^{i\varphi_{1}}\cdot e^{i\varphi_{3}}\right\}\cdot H_{3}\right\}\right|}^{2}$$
where $H_{3}$ is the transfer function from CL to QD target surface in COS-QD, corresponding to the $d_{3\;}$of Figure 6b. The distorted spots cause the QD detection to deviate from the center (0, 0), and the offset (i.e., the detection errors) obtained by solving $e_{\mathrm{NI}-\mathrm{MLA}}$ and $e_{\mathrm{COS}-\mathrm{QD}}$ using Equation (1) are $x_{{\mathrm{NI}-\mathrm{MLA}}_{\mathrm{error}}}$ and $x_{{\mathrm{COS}-\mathrm{QD}}_{\mathrm{error}}}$, respectively (taking the X direction as an example). In order to evaluate the mitigation effect of the NI-MLA system on the above errors, this paper uses the *RMSE* of the QD computation offset as the evaluation index and compares ${\mathrm{RMSE}}_{\mathrm{NI}-\mathrm{MLA}}$ and ${\mathrm{RMSE}}_{\mathrm{COS}-\mathrm{QD}}$ to characterize the improvement of the NI-MLA system over the COS-QD system in mitigating the SPDRE-QD under turbulent conditions.

Before the simulation parameters are given, the parameters of the two optical systems need to be constrained. In Equation (1), the coefficient *k* significantly affects the QD computation, and its value is related to the spot size [22]. In order to compare the SPDRE-QD of the two optical systems under the same conditions, the parameter constraints for the two optical systems have been established by matching the spots generated by both systems with QD without exceeding the QD’s detection range. As shown in Figure 7a,c, in the NI-MLA system when the diagonal length of the square spot is equal to the QD radius and in the COS-QD when the QD radius is twice the beam waist radius of the spot, both can achieve a balance between QD detection sensitivity and detection range (as shown in Figure 7b,d, the red line is the spot detection range, and the yellow shadow is the spot centroid detection range). Therefore, in the subsequent simulation and experiment, the relationship between the side length *L* of the square spot generated by the NI-MLA system and the Gaussian beam waist radius *ω* generated by the COS-QD is given by ω = *L*/$\sqrt{2}$. COS-QD, light source, and atmospheric parameters are listed in Table 3.

Based on the parameters in Table 1 and Table 3, as well as applying Equation (1) to solve $e_{\mathrm{COS}-\mathrm{QD}}$ and $e_{\mathrm{NI}-\mathrm{MLA}}$, and using $C_{n}^{\;2}=1\times 10^{-12}$ as a demonstration example, Figure 8a shows the QD computation errors of the two optical systems under strong turbulence conditions. It can be observed that the position distribution of the distorted spots modulated by the NI-MLA system and computed by the QD (orange and red scatter points) is more concentrated than the COS-QD (blue scatter points). Specifically, the *RMSE* of the QD computation values is calculated in the x-axis direction, ${\mathrm{RMSE}}_{\mathrm{NI}-\mathrm{MLA}}=0.0243\;\mathrm{mm}\;\mathrm{for}\Delta z=0$, ${\mathrm{RMSE}}_{\mathrm{NI}-\mathrm{MLA}}=0.0177\;\mathrm{mm}$ for Δz =1.5 mm, ${\mathrm{RMSE}}_{\mathrm{COS}-\mathrm{QD}}=0.121\;\mathrm{mm}$ (Corresponding to the orange, yellow, and blue bar charts in Figure 8b with a horizontal axis $C_{n}^{\;2}$ of $10^{\;-12}$, respectively). This simulation results indicate that under strong turbulence conditions, the NI-MLA system outperforms the COS-QD in mitigating the SPDRE-QD.

In Figure 8b, the left black vertical axis corresponds to the *RMSE* of the QD computation values for both the COS-QD and NI-MLA systems under different turbulence conditions, represented by blue, orange, and yellow bar charts. As for the right red vertical axis, ${E\;=\;\mathrm{RMSE}}_{\mathrm{COS}-\mathrm{QD}}/{\mathrm{RMSE}}_{\mathrm{NI}-\mathrm{MLA}}$ is defined as the evaluation index of relative mitigation effect for the NI-MLA system and is characterized by a line graph. The horizontal axis represents the atmospheric refractive index structure constant $C_{n}^{2}$ ranging from $1\times 10^{-16}$ to $1\times 10^{-12}{\;m}^{-2/3}$. The results indicate that, first, as the turbulence intensity increases from weak to strong, the NI-MLA system demonstrates a significant advantage in mitigating the SPDRE-QD compared with the COS-QD, and its relative mitigation effect *E* is relatively stable (when Δz =1.5 mm, *E* shows a slight downward trend, while at Δz=0, the trend is the opposite) under different turbulence intensity conditions. In addition, when the QD target surface is located at the defocus position of the NI-MLA system (Δz =1.5 mm), the *RMSE* of QD computation values are consistently lower than at the focal plane (Δz=0). This indicates that the capability of the NI-MLA system to mitigate the SPDRE-QD under atmospheric turbulence can be changed by adjusting the defocus amount to some extent. The discussion of the system’s capability of mitigation under more defocus parameters will be further elaborated in the Experimental Results section.

## 5. Experimental Results

### 5.1. Experimental Setup

As shown in Figure 9, an experimental system was designed to verify the AMC of the NI-MLA system and mitigate the SPDRE-QD under atmospheric turbulence. In the absence of turbulence, the system was used to verify the angular measurement characteristic and QD computation characteristic of the NI-MLA system (corresponding to Section 3). In the presence of turbulence, the performance of the NI-MLA system in mitigating the SPDRE-QD (corresponding to Section 4) and the impact of relative position ($d_{1},\Delta z$) change of MLA and QD on its mitigation effect were tested.

The diagram and photography of the experiment are shown in Figure 9. A laser with a wavelength of 632.8 nm is emitted from the collimator (f =7.93 mm, NA = 0.50 FC/PC) at a power of 400 μW, which maintains a high signal-to-noise ratio for the QD without saturation. After a segment of free-space transmission, the beam is expanded and polarized before being incident at a certain angle onto the center of the liquid crystal spatial light modulator (SLM, 1920 × 1080, 8 µm, 6π, 60 Hz). The phase modulation capability of this device is used to simulate the wavefront of the beam passing through atmospheric turbulence. The modulated laser beam is incident on the NI-MLA system, sequentially passing through the MLA (parameters as shown in Table 1) and a focusing lens (f=25 mm) and then focusing on the QD target surface (radius 5 mm, resolution 0.7 µm). The actual displacement of the spot is measured and recorded by the piezo stage (resolution 3 nm) below the QD. The output voltage from each quadrant of the QD is collected by an analog-to-digital converter and analyzed by a computer to give the computed position of the spot. In the NI-MLA system, CL and MLA are each fixed on five-axis adjustment stages to ensure accurate alignment of the optical components with QD. As shown in Figure 9b, the QD and NI-MLA system are placed as a whole on the azimuth rotation stage (represented by the pink block in Figure 9a), and the system’s rotation angle is accurately measured using a theodolite (resolution of 2.5 urad). In addition, in order to enhance the system’s flexibility and adjustability, the MLA can be laterally moved out of the optical path using a horizontal linear translation stage (represented by the green block in Figure 9a), allowing for easy conversion from the NI-MLA system to the COS-QD.

### 5.2. Experimental Verification of the AMC of the NI-MLA System

In the experiment to verify the AMC of the NI-MLA system, the plane wave with uniform light intensity distribution emitted by the interferometer with a wavelength of 632.8 nm is used to replace the Gaussian light emitted by the collimator in Figure 9 at the same position. The SLM is loaded with zero phase equivalent to a reflector, and the azimuth rotation stage below the QD and NI-MLA system is adjusted to change the incident angle of the beam $\theta_{i}$, which is accurately measured using a theodolite. At this point, the upper computer shows that the spot deviates from the QD center, and the piezo stage under the QD is used to adjust the spot until the spot returns to the QD center again, and the displacement data ${\Delta y}_{o\;}$(spot centroid displacement) are recorded. Same as in the simulation, in the experiment, the $\theta_{i}$ in the range of ±500 µrad (in 50 µrad increments) is chosen for testing. The NI-MLA system parameters are based on Table 1, and the goodness of fit (*R*^2^) is also used as an evaluation index to verify the AMC of the NI-MLA system.

Figure 10a,b shows the relationship curves between ${\mathrm{EC}}_{x}$ and $\theta_{i}$ for the NI-MLA system at different defocus positions. The green lines are obtained by the first-order linear fitting of the measured value of ${\mathrm{EC}}_{x}$ (blue scatter points). The goodness of fit *R*^2^ is greater than 0.99 when the $\theta_{i}$ changes in the range of 1 mrad (due to the measurement error of experimental instruments such as theodolite and the manufacturing error of experimental devices such as the NI-MLA system, the *R*^2^ in the experiment is lower than the simulated value). The above results show that within ±500 µrad, there is a linear relationship between${\;\mathrm{EC}}_{x}\;$and${\;\theta }_{i}$, demonstrating that the NI-MLA system possesses the AMC. As shown in Figure 10c, the variation in the focal length of the NI-MLA system is investigated when the QD target surface is at different defocus positions, and the Δz range is set from −2 mm to 2 mm, in an increment of 0.5 mm. Other parameters are listed in Table 1. The theoretical values, simulated values, and experimental measurements of the NI-MLA system’s focal length are represented by the red line, blue circles, and green circles in Figure 10c, respectively. It can be observed that although the experimental measurements of the system’s focal length are closer to the theoretical values than the simulated values in some defocus, the simulations show the same trend as the experimental measurement. This indicates that compared with the theoretical values obtained by geometric optics, the focal length simulated results obtained by Fourier optics considering the MLA aperture diffraction effect are closer to the measured focal length of the NI-MLA system in the change trend, which reflects the accuracy of the simulation model in simulating the NI-MLA system. At the same time, it can be observed that when ∆*z* < 0, the difference between the theoretical, simulated, and experimental values is relatively large. However, when ∆*z* > 0, the difference is smaller. This discrepancy can be attributed to the difference in spot uniformity when the NI-MLA system focuses in front of or behind the focal plane. When the focus is in front (∆*z* < 0), the spot uniformity is lower compared with focusing behind (∆*z* > 0), resulting in a larger deviation in the spot’s centroid position when the incident light is tilted at a certain angle.

In summary, it has been collectively verified that the NI-MLA system possesses the AMC within the paraxial region (±500 μrad) by exploring the relationship between the spot energy centroid position${\;\mathrm{EC}}_{x}$ and the beam incident angle $\theta_{i}$, as well as measuring the system focal length at different defocus positions. However, QD is highly sensitive to the uniformity of the spot energy distribution when computing the spot position. Therefore, when $\theta_{i}$ ranges linearly, the linear displacement of the emitted spot on the QD target surface does not mean that the spot position computed by QD will exhibit the same linear displacement. Further investigation is needed to explore the QD computation curve characteristics at different parameter settings.

### 5.3. QD Computation Curve Characteristics

As shown in Figure 11, under the same input parameter conditions as in Figure 4 of Section 3.2 (see Table 1), the experimental platform illustrated in Figure 9, equipped with the QD and NI-MLA system is used. And the SLM is loaded with zero phase equivalent to a reflector. This section experimentally measures and plots the QD computation values $\sigma_{x}$ of the spot displacement and the true value *X* of the spot displacement (piezo stage moving step of 20 µm, with a total of 101 test points) for the NI-MLA system, represented by the blue curves. Additionally, the computation curves for the COS-QD are measured as reference, represented by the red curve. In Table 4, the QD computation curve characteristics (linearity $L_{x}$, sensitivity ${\;k}_{x}$, and detection accuracy ${\mathrm{RMSE}}_{x}$) of the NI-MLA system and COS-QD at different defocus positions (∆*z* = −2 mm to 2 mm, step is 0.5 mm for the NI-MLA system) are experimentally explored. The two optical systems are as shown in Figure 6, and the relationship between the defocus amount Δz of the NI-MLA system and the defocus amount $\Delta z_{1}$ of the COS-QD is established by ω = *L*/$\;\sqrt{2}$ allowing us to effectively compare the differences in QD computation characteristics of the two optical systems at various defocus positions.

In Figure 11, the curve characteristics of the NI-MLA system (corresponding to${\;L}_{x}$, $k_{x}$ and ${\mathrm{RMSE}}_{x}$ values at Δz =0 and Δz =1.5 mm in Table 4) align with the simulation results (Table 2) in parts (iii) of Figure 4a,b, which confirms the conclusion in Section 3.2 that “In the NI-MLA system, the QD computation curve characteristics can be optimized by appropriate defocus”. As for the QD computation curves and their characteristics of the COS-QD (corresponding to $L_{x}$, ${\;k}_{x}$ and ${\mathrm{RMSE}}_{x}$ values at $\Delta z_{1}=-5.30\;\mathrm{mm}$ and $\Delta z_{1}=-4.92\;\mathrm{mm}$ in Table 4), it can be observed that at Δz =0 and $\Delta z_{1}=-5.30\;\mathrm{mm}$, the $L_{x}$ and ${\mathrm{RMSE}}_{x}$ of the NI-MLA system are inferior to those of the COS-QD, while at the defocus of the NI-MLA system, that is, Δz =1.5 mm and $\Delta z_{1}=-4.92\;\mathrm{mm}$, the $L_{x}$, $k_{x}$, and ${\mathrm{RMSE}}_{x}$ of the NI-MLA system are superior to those of the COS-QD.

In Table 4, it can be observed that linearity $L_{x}$ and detection accuracy ${\mathrm{RMSE}}_{x}$ of the NI-MLA system reach the lowest level at Δz=0 mm, and once the QD target surface is at defocus position of the NI-MLA system, both $L_{x}$ and${\;\mathrm{RMSE}}_{x\;}$improve. Specifically, when Δz < 0, as the defocus amount of the NI-MLA system decreases, the diffraction effects become more pronounced, and the linearity and detection accuracy of the computation curves gradually decline. Compared with the NI-MLA system, the COS-QD performs better in terms of linearity and detection accuracy. When Δz > 0, as the defocus amount increases, the reduction in diffraction effects leads to an increase in spot uniformity, resulting in a significant improvement in the linearity and detection accuracy of the NI-MLA system, with the linearity and detection accuracy of the NI-MLA system surpassing those of the COS-QD. Especially at Δz=2 mm, the NI-MLA system exhibits the highest linearity, sensitivity, and detection accuracy.

Therefore, these experimental results verify that the QD computation curve characteristics of the NI-MLA system can be significantly optimized by properly defocusing the QD target surface. Additionally, with the appropriate parameter selection, the QD computation curve characteristics of the NI-MLA system can comprehensively outperform those of the COS-QD.

### 5.4. Turbulence Mitigation Effect of NI-MLA System

This subsection has verified the ability of the NI-MLA system to mitigate the SPDRE-QD under turbulence conditions and examined the influence of different system parameters on this mitigation effect. The SLM is loaded with the phase distribution that conforms to the Kolmogorov power spectrum. After being emitted from the collimator, the laser is reflected by the SLM, effectively simulating the distorted spot produced by 1 km laser propagation under different turbulence intensities in Table 3 (with each $C_{n}^{2}$ corresponding to 10,000 different shapes of distorted spots). Figure 12a shows images captured by replacing the QD target surface with a camera in both optical systems (parameters are shown in Table 1 and Table 3). By comparing the spot morphology of the two systems with or without turbulence input, it can be qualitatively observed that the NI-MLA system, after loading atmospheric turbulence (represented as AT in the figure), exhibits a significantly improved stability in spot energy distribution compared with the COS-QD.

In Figure 12b, the experimental measurements show the spot position distribution of QD computation for both optical systems under strong turbulence conditions ($C_{n}^{2}=1\;\times \;10^{-12}$), as well as the detection errors and the relative mitigation effect *E* of the NI-MLA system under weak, medium, and strong turbulence conditions. First, the experimental results of the COS-QD (blue scatter points in part (i) and blue bars in part (ii) of Figure 12b) are compared with the simulation results (blue scatter points in Figure 8a and blue bars in Figure 8b). The two are consistent in the magnitude of QD calculation errors, verifying the effectiveness of using SLM for turbulence simulation in the experiment. Second, it can be observed that when utilizing the NI-MLA system, the *RMSE* of QD detection is significantly reduced compared with the COS-QD. This experimental phenomenon confirms the key conclusion of this study: the NI-MLA system has the characteristics to mitigate SPDRE-QD under various turbulence conditions. However, by comparing the simulation results (orange and yellow scatter points in Figure 8a and orange and yellow bars in Figure 8b) with the experimental results (orange and yellow scatter points in part (i) and orange and yellow bars in part (ii) of Figure 12b), it is observed that the position distributions of the spot are more concentrated (smaller *RMSE*) in the simulation, that is, a better relative mitigation effect *E* in the simulation (compare the line chart in Figure 8b with the line chart in part (ii) of Figure 12b). The analysis shows that, in the comparison of the above simulation and experimental phenomena, the simulation can predict the general trend of the relative mitigation effect *E* of the NI-MLA system, and the maximum error is no more than 18%. The error can be attributed to the loss of some details of intensity variation when using the discretized grid matrix to characterize the spot in the simulation, which makes the spot energy distribution in the simulation more uniform than the experimental results, resulting in an underestimation of the QD computation errors for the NI-MLA system in the simulation. On the other hand, factors such as the randomness of turbulence simulated by SLM, the phase modulation capability of the SLM, the consistency of the four quadrants of the QD, the surface accuracy of MLA, and the coaxial alignment of the optical system in the experiment can diminish the homogenization effect on the spot, and then the QD computation errors of the NI-MLA system in the experiment are slightly larger, resulting in the relative mitigation effect *E* lower than the simulation results.

## 6. Conclusions

We have presented analytical and experimental results for our proposed NI-MLA application in QD optical systems. The results demonstrate that the NI-MLA system can reduce the impact of atmospheric turbulence on QD computations, and the maximum detection error can be mitigated by 17.36 times in the experiment. A key feature of the NI-MLA is the angular measurement characteristic, which is proved theoretically by geometric optics for the first time and is the prerequisite for its use as a QD optical system. The study also emphasizes that the parameter selection for the NI-MLA system must comprehensively consider multiple indicators, such as the mitigation effect on QD detection errors and QD computation curve characteristics. When the QD target surface is in the positive defocus position (Δz > 0) of the NI-MLA system, it outperforms the COS-QD in terms of turbulence mitigation capability and, also, the detection linearity and accuracy. In conclusion, this study demonstrates the potential of the NI-MLA system not only to enhance QD detection performance in laser communication and laser guidance systems under atmospheric turbulence but also to reduce pointing errors, which, in turn, enhances the tracking stability of the system. In addition, it provides significant contributions to the application of MLA technology, broadening its application prospects in high-precision measurement fields.

## Figures and Tables

**Figure 1 sensors-24-06684-f001:**
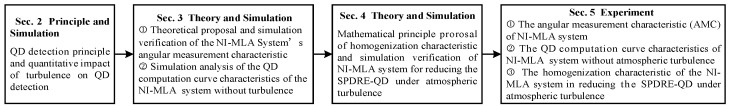
The outline block diagram of this paper.

**Figure 2 sensors-24-06684-f002:**
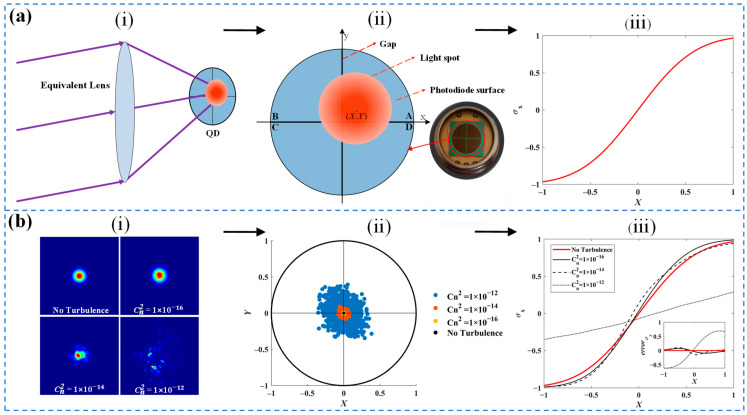
Measurement principles of the quadrant detector and computational characteristics of spot distortion under atmospheric turbulence. (**a**) Measurement principles of the quadrant detector. (**i**) The principle of spot detection in the quadrant detector. (**ii**) Schematic diagram of the four-quadrant detector structure. (**iii**) The relationship between $\sigma_{x}$ and X. (**b**) QD computation characteristics of the COS-QD under varying intensities of atmospheric turbulence. (**i**) Several forms of spot distortion. (**ii**) Distorted spot position distribution by QD computation. (**iii**) QD computation curves corresponding to the distorted spot.

**Figure 3 sensors-24-06684-f003:**
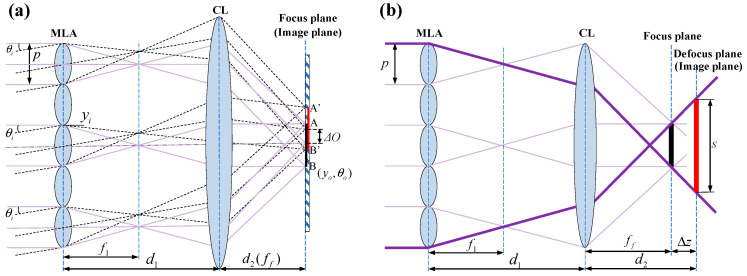
Schematic diagram of NI-MLA system. (**a**) The image plane (QD target surface) is located at the focal plane of CL. (**b**) The image plane is located at the defocus of CL.

**Figure 4 sensors-24-06684-f004:**
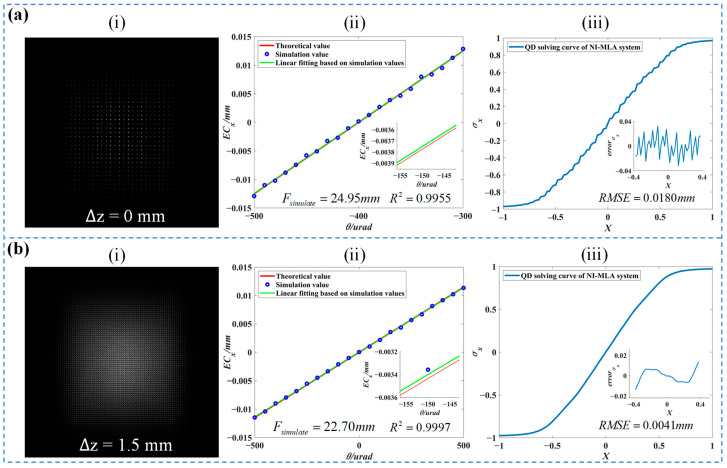
Simulation of the AMC and QD computation curve of the NI-MLA System. (**a**) Part (**i**) represents the spot image on the QD target surface when Δ*z* = 0. Part (**ii**) represents the relationship between ${\mathrm{EC}}_{x}$ and θ, calculated from the spot energy distributions shown in part (**i**) using Equation (6) when Δ*z* = 0. Part (**iii**) represents the QD computation curves corresponding to the spots in part (**i**), calculated using Equation (1) when Δ*z* = 0. (**b**) Part (**i**) represents the spot image on the QD target surface when Δ*z* = 1.5 mm. Part (**ii**) represents the relationship between ${\mathrm{EC}}_{x}$ and θ, calculated from the spot energy distributions shown in part (**i**) using Equation (6) when Δ*z* = 1.5 mm. Part (**iii**) represents the QD computation curves corresponding to the spots in part (**i**), calculated using Equation (1) when Δ*z* = 1.5 mm.

**Figure 5 sensors-24-06684-f005:**
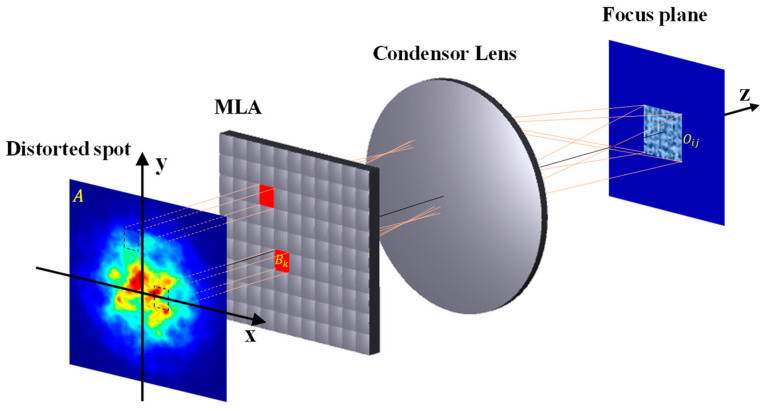
Schematic diagram of the NI-MLA system homogenization and shaping.

**Figure 6 sensors-24-06684-f006:**
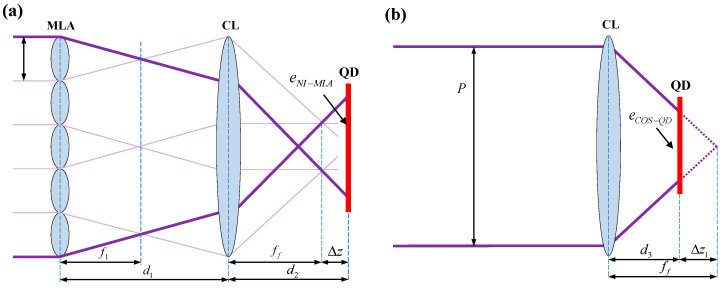
(**a**) QD optical system based on the NI-MLA system. (**b**) Conventional optical system of the QD (COS-QD).

**Figure 7 sensors-24-06684-f007:**
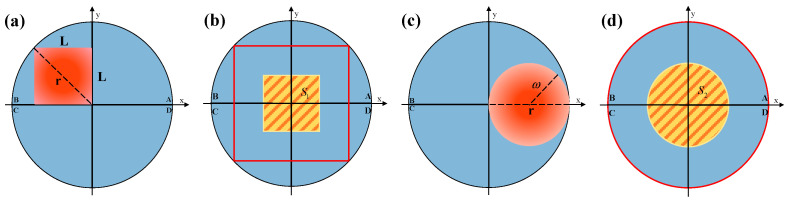
The best matching scheme for spots generated by NI-MLA and COS-QD with the QD. (**a**) The size relationship between the square Spot and QD. (**b**) The computation range of the square spot on QD. (**c**) The size relationship between the circle spot and QD. (**d**) The computation range of the circle spot on QD.

**Figure 8 sensors-24-06684-f008:**
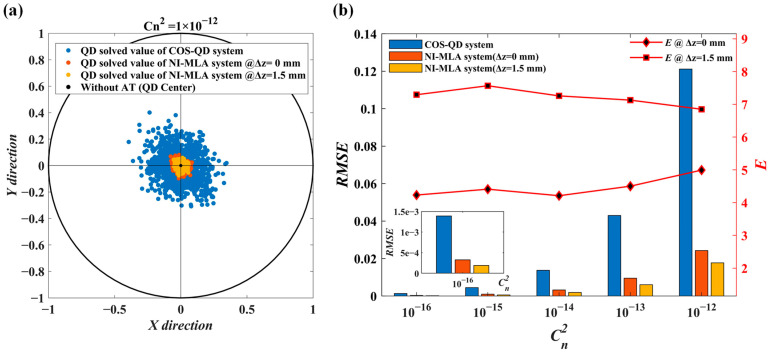
Comparison diagram of QD detection error simulation between NI-MLA and COS-QD in turbulence. (**a**) The QD detection error for 10,000 sets of distorted spots passing through the two optical systems at $C_{n}^{\;2}=1\times 10^{\;-12}$. (**b**) The *RMSE* of the QD computation values for the two optical systems and the relative mitigation effect *E* of the NI-MLA system.

**Figure 9 sensors-24-06684-f009:**
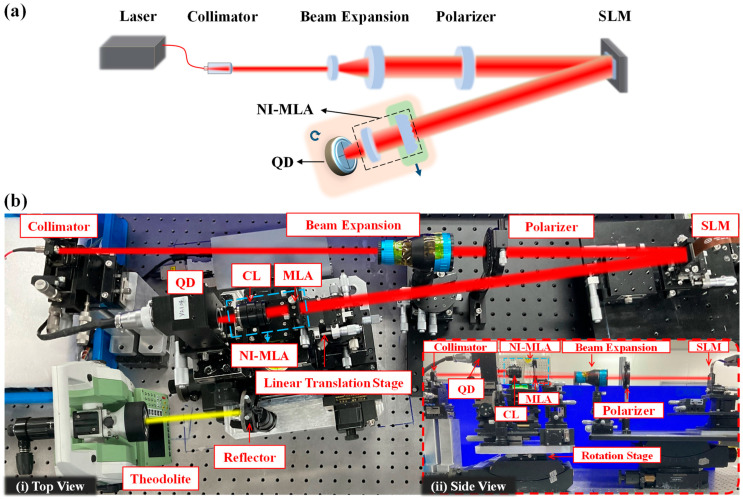
Diagram and photograph of the experiment. (**a**) Diagram of the experimental setup. Pink block: high-precision rotation stage. Green block: linear translation stage. (**b**) Photograph of the experiment. (**i**) Top view of the optical path diagram. (**ii**) Side view of the optical path diagram.

**Figure 10 sensors-24-06684-f010:**
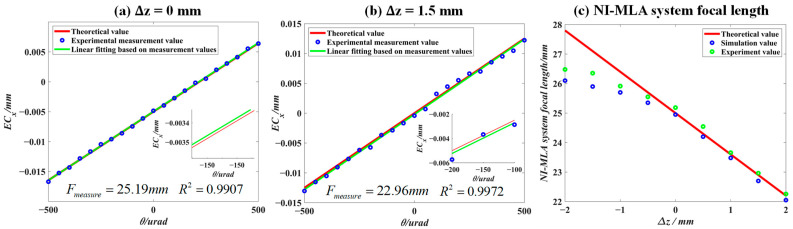
Experimental verification of the AMC of the NI-MLA system. (**a**) Measurement of the relationship between the spot energy centroid position ${\mathrm{EC}}_{x}$ and incident angle $\theta_{i}$ for Δz=0. (**b**) Measurement of the relationship between spot centroid displacement ${\Delta y}_{o}$ and incident angle $\theta_{i}$ for Δz=1.5 mm. (**c**) The theoretical, simulated, and experimental focal length values of the NI-MLA system for Δz=−2 to 2 mm.

**Figure 11 sensors-24-06684-f011:**
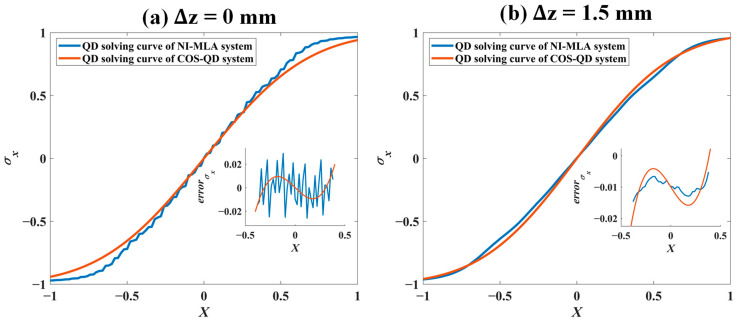
QD computation curve between tghe NI-MLA system and COS-QD.

**Figure 12 sensors-24-06684-f012:**
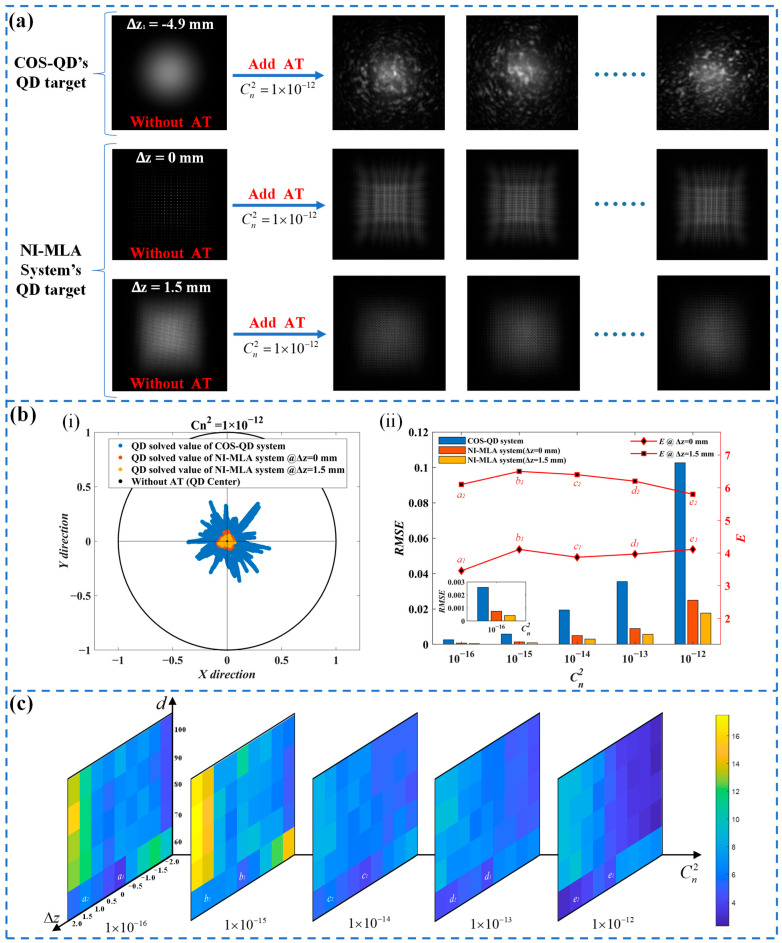
Turbulence mitigation effect of the NI-MLA system. (**a**) Spot distribution images on the QD target surface of the COS-QD and NI-MLA systems before and after loading turbulence (captured by a camera at the corresponding position). (**b**) Part (**i**) shows the spot position distribution obtained from QD computation of the spots shown in figure (**a**) when $C_{n}^{2}=1\;\times \;10^{-12}$. Part (**ii**) shows the *RMSE* of QD computation values and the relative mitigation effect E for the COS-QD and NI-MLA systems at different defocus positions under various turbulence conditions. (**c**) The color scale diagram of the relative mitigation effect E of the NI-MLA system with different MLA and QD positions under various turbulence conditions, where figure (**c**) corresponds to *a*_1_ to *e*_2_ in part (**ii**) of figure (**b**).

**Table 1 sensors-24-06684-t001:** NI-MLA system parameters.

Parameter Name	Symbol	Value
MLA sub-aperture size	*p*	300 μm
MLA sub-aperture focal length	*f* _1_	5 mm
Number of MLA sub-aperture	*a*	50 × 50
The focal length of the condenser lens	*f_f_*	25 mm
Distance between MLA and CL	*d* _1_	60 mm
QD target surface defocus	Δz	0 and 1.5 mm

**Table 2 sensors-24-06684-t002:** NI-MLA system’s simulation results of AMC and QD computation curve characteristics.

Category	Parameter	Δz =0 mm	Δz =1.5 mm
NI-MLA system’s AMC	*R* ^2^	0.9955	0.9997
	$F_{\mathrm{simulate}}$/$F_{\mathrm{theory}}$/mm	24.95/25.00 mm	22.70/22.90 mm
NI-MLA system’s QD computation curve characteristics	$L_{x}$	0.0261	0.0112
	$k_{x}$	1.4205	1.5733
	${\mathrm{RMSE}}_{x}$/mm	0.0180 mm	0.0041 mm

**Table 3 sensors-24-06684-t003:** COS-QD, light source and atmospheric parameters.

Category	Parameter Name	Symbol	Value
COS-QD system	focal length	$f_{f}$	25 mm
	effective aperture	*P*	10 mm
	defocus	$\Delta z_{1}$	−4.92 mm
Light source	wavelength	*λ*	632.8 nm
	waist radius	$\omega_{1}$	1.2 mm
Atmospheric parameters	refractive index structure constant	$C_{n}^{2}$	$1\times 10^{-16}\;\sim \;1\times 10^{-12}{\;m}^{-2/3}$
	atmospheric channel length	*D*	1 km

**Table 4 sensors-24-06684-t004:** QD computation curve characteristics of the NI-MLA system and COS-QD.

Δz /mm	$\Delta z_{1}$/mm	$L_{x}$		$k_{x}$		${\mathrm{RMSE}}_{x}$/mm	
		COS-QD	NI-MLA	COS-QD	NI-MLA	COS-QD	NI-MLA
−2.00	−5.81	0.0155	0.0194	1.3727	1.4908	0.0043	0.0061
−1.50	−5.69	0.0162	0.0205	1.4035	1.5143	0.0046	0.0068
−1.00	−5.56	0.0170	0.0211	1.4356	1.5533	0.0048	0.0071
−0.50	−5.43	0.0177	0.0224	1.4692	1.6015	0.0050	0.0079
0.00	−5.30	0.0186	0.0274	1.5045	1.6309	0.0052	0.0171
0.50	−5.17	0.0195	0.0145	1.5415	1.6818	0.0055	0.0045
1.00	−5.05	0.0204	0.0139	1.5804	1.7179	0.0057	0.0036
1.50	−4.92	0.0215	0.0123	1.6212	1.7509	0.0060	0.0033
2.00	−4.79	0.0226	0.0105	1.6643	1.8157	0.0063	0.0028

## Data Availability

The raw data supporting the conclusions of this article will be made available by the authors on request.

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
