# Peer review of "Research on the NI-MLA Method for Enhancing the Spot Position Detection Accuracy of Quadrant Detectors Under Atmospheric Turbulence"

_sensors, 2024, doi:10.3390/s24206684_

Round 1

Reviewer 1 Report

Comments and Suggestions for Authors

The authors propose and demonstrate an attractive technique to improve the accuracy of quadrant detector under atmospheric turbulence.I believe the authors obtained the valuable results.

The paper could be accepted after minor revision. In particular it would be helpful  for better clarity to explain in a liittle more details the results presented on Fig.2a and Fig.2b

Comments on the Quality of English Language

I am not perfectly certain but probably there are too much "comma" in text.

Reviewer 2 Report

Comments and Suggestions for Authors

more details seen in the attachment file

Reviewer 3 Report

Comments and Suggestions for Authors

The article adopts the NI-MLA method to improve the accuracy of detecting the spot position of the four quadrant detector under atmospheric turbulence, with detailed content and experimental verification. Please add the following questions before the paper can be accepted:

The light spot formed in Figure 2 appears to have been generated under strong turbulence, please note its consistency with the subsequent experimental conditions.

Has the author considered the impact of flickering light intensity on detection accuracy?
